# ALS2-Related Motor Neuron Diseases: From Symptoms to Molecules

**DOI:** 10.3390/biology11010077

**Published:** 2022-01-05

**Authors:** Marcello Miceli, Cécile Exertier, Marco Cavaglià, Elena Gugole, Marta Boccardo, Rossana Rita Casaluci, Noemi Ceccarelli, Alessandra De Maio, Beatrice Vallone, Marco A. Deriu

**Affiliations:** 1PolitoBIOMedLab, Department of Mechanical and Aerospace Engineering, Politecnico di Torino, 10129 Torino, Italy; marcello.miceli@polito.it (M.M.); marco.cavaglia@polito.it (M.C.); marta.boccardo@studenti.polito.it (M.B.); rossanarita.casaluci@studenti.polito.it (R.R.C.); noemi.ceccarelli@studenti.polito.it (N.C.); a.demaio@studenti.unipi.it (A.D.M.); 2Dipartimento di Scienze Biochimiche “A. Rossi Fanelli”, Sapienza Università di Roma, 00185 Rome, Italy; cecile.exertier@uniroma1.it (C.E.); elena.gugole@uniroma1.it (E.G.); beatrice.vallone@uniroma1.it (B.V.)

**Keywords:** Alsin, IAHSP, JPLS, JALS, rare diseases, protein, mutations, neurodegenerative

## Abstract

**Simple Summary:**

Mutations of the ALS2 gene, which encodes for the protein Alsin, are linked to three recessive motor neuron diseases characterized by early onset. Alsin is an intriguing protein characterized by several structured domains with distinct functions. To date, it is not fully understood how the aforementioned domains collaborate in the development of Alsin functions and how mutations, located in specific areas of these domains, correlate with Alsin malfunction and disease onset. This study collects information from the literature rationalized on three levels of investigation: a systemic scale (symptoms of the pathology), a protein scale (molecular phenomena that drive the development of the pathology) and a population scale (comparison between ALS2-related diseases and detected mutations). Differences and similarities among ALS2-related diseases are comprehensively highlighted here and correlated with Alsin mutations.

**Abstract:**

Infantile-onset Ascending Hereditary Spastic Paralysis, Juvenile Primary Lateral Sclerosis and Juvenile Amyotrophic Lateral Sclerosis are all motor neuron diseases related to mutations on the ALS2 gene, encoding for a 1657 amino acids protein named Alsin. This ~185 kDa multi-domain protein is ubiquitously expressed in various human tissues, mostly in the brain and the spinal cord. Several investigations have indicated how mutations within Alsin’s structured domains may be responsible for the alteration of Alsin’s native oligomerization state or Alsin’s propensity to interact with protein partners. In this review paper, we propose a description of differences and similarities characterizing the above-mentioned ALS2-related rare neurodegenerative disorders, pointing attention to the effects of ALS2 mutation from molecule to organ and at the system level. Known cases were collected through a literature review and rationalized to deeply elucidate the neurodegenerative clinical outcomes as consequences of ALS2 mutations.

## 1. Introduction

Infantile-onset ascending hereditary spastic paralysis (IAHSP), Juvenile Primary Lateral Sclerosis (JPLS), and Juvenile Amyotrophic Lateral Sclerosis (JALS) are motor neuron diseases (MNDs) characterized by isolated pyramidal cells’ degeneration. Clinical manifestations appear from the first years of life with a progressive lower-limb spasticity that, over the years, reaches the upper limbs, leading to a quadriplegia condition [1,2,3,4,5,6,7,8,9,10]. A common molecular feature may characterize the above-mentioned pathologies, i.e., mutations in the Amyotrophic Lateral Sclerosis type 2 (ALS2) gene, encoding for Alsin, a structured protein, and a player of essential roles in the cell, including acting as GTPase regulator and a controller of the survival and growth of spinal motoneurons [4,10,11,12,13]. Alsin mutations would be responsible for the altered behavior of the protein’s native oligomerization state or a change in Alsin’s propensity to interact with protein partners, even altering endosomal function. The aforementioned pathologies have been studied in the last decades using unconnected but complementary strategies that have included clinical, biological, and molecular investigations. Despite the advances in the knowledge on this disease, to date, there are essentially no substantial cures that stop the degenerative nature of here-considered MNDs. One of the limits of the present scientific research could be related to the fact that the structure of Alsin has not been resolved yet. This aspect limits the understanding of the molecular mechanisms that are certainly related to the misfunctioning of Alsin as a result of mutations that, most likely, lead to distortions of the protein’s tertiary structure and consequent alterations in function, including modifications of the associated protein cascades that subsequently impact growth and correct the development of motor neurons. A second limitation is that we are facing very rare pathologies, which reduce the available amount of clinical data that is necessary for achieving a satisfactory robustness of models and results. Finally, the research in the field of these diseases has so far been mainly sectorial, focusing on the patient at the clinical level and on a specific protein function at the cellular level. In essence, the lack of knowledge at the molecular level, the limited number of patients, and the lack of research projects involving multidisciplinary networks constrain our ability to obtain a comprehensive view of the disease and how, from specific molecular events, the pathology evolves. Within this vision is placed this review work, which proposes a comparison of IAHSP, JPLS, and JALS, making use of different levels of pathology description. At the macroscopic level, i.e., from a systems perspective, the symptoms and clinical features that characterize the diseases are compared. At the microscopic level, the molecular features of IAHSP, JPLS, and JALS are highlighted, focusing on Alsin and its role in cellular functions, both in physiological and pathological conditions. Finally, starting from an overview that integrates pathological microscopic features and clinical symptoms, a population-level view is given, summarizing the known cases of IAHSP, JALS, and JPLS reported in the literature. Particular attention will be paid to the region of the genome involving the mutation, the type of mutation, and the clinical characteristics of the patients. In conclusion, this work represents a first attempt to rationally bring together the available relevant knowledge from an interdisciplinary collection of studies investigating ALS2-related diseases at different scales, and it attempts to answer to the need for an open-access research framework giving attention to rare neurological conditions such as IAHSP (prevalence < 1:100,000), where scientific advancements are limited by the quantity, availability, heterogeneity, dispersion, and fragmentation of patient data.

## 2. Macroscopic Level: Clinical Features of IAHSP and Other ALS2-Related Pathologies

Recessive mutations in the ALS2 gene are responsible for distinct MND conditions, namely Infantile-onset ascending hereditary spastic paralysis (IAHSP, OMIM:607225), Juvenile Primary Lateral Sclerosis (JPLS, OMIM:606353), and Juvenile Amyotrophic Lateral Sclerosis (JALS OMIM:205100) [3,14]. The main features characterizing above-mentioned diseases are listed in Table 1 and detailed in the following.

IAHSP is caused by a mutation in the ALS2 gene, locus 2q33.1, encoding for the Alsin protein [4,10,11,12]. IAHSP was classified as a pure form of Hereditary Spastic Paraplegia (HSP) [12]; this disease is inherited in an autosomal-recessive manner and presents with isolated pyramidal degenerative signs [5,6,7,8,9,10]. Symptoms appear during the first years of life, manifesting as a spasticity initially involving the lower limbs and progressing over the next ten years to affect the upper limbs, leading to quadriplegia. [1,2,3,4]. The symptomatology would appear to occur as a consequence of retrograde degeneration of the upper motor neurons of the pyramidal tracts induced by a mutation in the ALS2 gene. By the age of ten, these patients are wheelchair-dependent and in the following decade of life, the disease tends to progress toward severe spastic tetraparesis and pseudobulbar syndrome (anarthria and dysphagia), requiring a gastrostomy tube [2,4].

JPLS is a rare infantile-onset neurodegenerative disease that begins during the first years of life, between 1 and 3 years [3]. The disease is characterized by autosomal-recessive hereditary transmission, caused by mutations in the ALS2 gene, unlike the adult primary lateral sclerosis, which is inherited in an autosomal-dominant manner [3,15,16,17,18]. Symptoms of JPLS appear during the first years of life and progress over a period of 15 to 20 years. Clinically, JPLS is very similar to IAHSP in that shows progressive signs of upper motor neuron degeneration leading to wheelchair dependence by adolescence and, later, to motor speech impairment [3,17,19,20]. A peculiar symptom of JPLS is the diffuse conjugate saccadic gaze paresis (uncontrolled eye movements), which is particularly severe upon downgaze (eyes looking downward) [3,15,17,19,20,21,22,23,24]. Survival of patients affected with this disease is variable and as with IAHSP patients, cognitive functions are preserved [3,17,19,20].

JALS is characterized by an onset during childhood, differently from the adult form (age at onset 58–63) or Sporadic forms of Amyotrophic Lateral Sclerosis (age at onset 58–63). This disease has been associated with mutations in different genes, i.e., ALS2, SPG11, SIGMAR1, SETX, UBQLN2, and FUS [25,26,27,28,29], with the last reported in the majority of cases [28]. JALS is inherited in an autosomal-recessive pattern involving *ALS2*, *SPG11*, and *SIGMAR1*, while the *SETX,* UBQLN2, and FUS mutations are described as an underlying cause of an autosomal-dominant form of JALS [28]. Clinical manifestation reflects the degeneration of both upper motor neurons and lower motor neurons causing weakness of the lower limbs, spasticity of facial muscles, uncontrolled laughter, dysarthria, bladder dysfunction, sensory disturbances, and, in rare cases, mental retardation and scoliosis [29,30]. Death usually occurs from respiratory failure between 7 and 84 months after onset [25,28,30,31,32,33,34,35]. Reported cases of JALS linked to a mutation in the ALS2 gene express a different phenotype than those linked to mutations in other genes. More specifically, symptoms generally appear later for ALS2-induced JALS patients (median age 4, 5 years), and the clinical course is slower [12,20,21,23,36].

To summarize, IAHSP, JPLS, and JALS are three diseases that can be related to mutations in the ALS2 gene and that share common symptoms, such as spasticity and weakness in the lower limbs, bulbar involvement, anarthria, dysphagia, isolated pyramidal signs, loss of sensation and control of the sphincters, and often loss of motor skills [1,2,3,4,17,19,20,29,30]. However these pathologies differ in terms of the age of onset (Table 1) [1,2,3,4,12,17,19,20,21,23,25,26,30,36,37] and the involvement of lower motor neurons, since this has only been described in IAHSP and JALS, but not in JPLS [3,15,20,30]. Finally, in terms of life expectancy, JALS seems to show the most severe phenotype compared with IAHSP and JPLS. Specifically, patients with JALS show a shorter lifespan, often due to respiratory failure (Table 1) [3,4,17,19,20,25,28,31,32,33,34,35].

## 3. Microscopic Level: Molecular Features of IAHSP and Other Related ALS2 Pathologies

The clinical course of the ALS2-induced MND forms has highlighted the need to investigate the molecular basis underlying these pathologies. At the lowest scale of investigation, attention is focused on the ALS2 gene, located on the long arm of chromosome 2q33 and composed of 34 exons. It is transcribed into two spliced transcripts, a long form with 1657 amino acids and a short form with 396 amino acids (Figure 1). Both proteins are expressed in various human tissues, but mostly in the brain (cerebellum) and spinal cord. However, some ectopic expression has been detected in the testis [11]. The larger transcript encodes for a protein called Alsin (molecular weight is 184 kD), localized on the cell cytoplasm, onto the cytoplasmic face of perinuclear and enlarged endosomes. Although the function(s) of Alsin are not yet fully elucidated, experimental evidence supports a role for Alsin as a guanine exchange factor (GEF) on small guanosine triphosphatases (GTPases). GEFs are known to activate GTPases by stimulating GDP dissociation and GTP binding, thereby activating the GTPases, which act as switches in numerous signaling cascades. Previous studies have demonstrated that Alsin preferentially targets the Rab5 GTPase family, which is involved in signal transduction, trafficking, and vesicle formation, suggesting that Alsin may play a key role in endocytosis and cytoskeletal reorganization in mammalian neuronal cells [4,24,38]. Alterations in the Alsin protein, ranging from point mutations to complete deletion of structured domains, lead to diverse clinical scenarios in terms of onset, symptoms, and fatality. Therefore, the description of Alsin’s structure and its involvement in protein cascades related to cellular functions through protein–protein interactions is an essential topic that must be understood in depth for a rational view of the mechanisms behind the development of the above mentioned MNDs.

### 3.1. Alsin Molecular Structure

Alsin is a protein constituted by three putative GEF domains [2,4,21]: the RCC1-like domain (RLD) at the N-terminus, the central B cell lymphoma (Dbl) homology (DH) and pleckstrin-homology (PH) domain, and the C-terminal vacuolar protein-sorting 9 (VPS9) domain. Additionally, eight consecutive membrane occupation and recognition nexus (MORN) motifs are inserted in between the DH/PH and VPS9 domains [4,39] (Figure 1).

Interestingly, the RLD, DH/PH, and VPS9 domains have shown GEF activity, supporting the role of Alsin as a regulator and activator of multiple small GTPases [38,40]. Moreover, Alsin’s structured domains seem to drive the subcellular localization of protein. More specifically, studies on truncated variants revealed that C-terminal-truncated variants are mainly cytoplasmic, while N-terminal-truncated variants display endosomal localization [41,42].

#### 3.1.1. RLD Domain and the Intrinsically Disordered Domain

Alsin’s N-terminal RCC1-like domain (RLD) shares homology with the RCC1 protein (Regulator of Chromosome Condensation 1-protein) [43,44]. Although clear experimental evidence is still lacking, homology modeling suggested that the Alsin RLD domain builds up a seven-bladed propeller by means of RCC1-like sequences of ~50-60 folded residues (propeller blades), which are stabilized by inter-blade disulphide bridges (Figure 1). Each blade of the propeller is composed of 4 β-strands connected in tandem by long loops: the N-terminal strand faces the central propeller tunnel, the second strand from one blade interacts with the third strand from the adjacent blade, and the fourth and shorter strand lies on the propeller surface. A conserved VyxWGT consensus sequence is observed on the third blade, which seems to participate in hydrophobic interactions between the blades, while the strong sequence variability of the last strand suggests that it could be involved in protein–protein interactions. It is worth noting that in the first blade, the first two beta-strands are constituted by the C-terminal part of the RLD domain, while the third and fourth strands are formed by the domain’s N-terminal sequence. This structural peculiarity was hypothesized to facilitate the completion of the propeller structure, favoring a correct folding of the domain owing to the first beta-strands from blade 1 acting as a velcro with the last two strands. Furthermore, in human Alsin, a ~300 amino acid sequence, predicted to be an intrinsically disordered region (IDR) and not evolutionarily conserved, segments the RCC1-like repeats in two regions and is inserted between the first two strands and the last two strands of blade 5. Disordered insertion in eucaryotic proteins is common and a structural transition may occur upon ligand/protein partner binding, as observed for many cellular signaling processes [45,46]. Literature studies suggested that the Alsin IDR may have a fundamental role in the proper self-oligomerization and in the intracellular distribution of the protein [47]. RCC1 is present in various proteins as a GEF for Ran (Ras-related nuclear), a small GTP-binding protein that is predominantly located in the nucleus and implicated in the nuclear import of proteins with nuclear localization signals and chromatin condensation through the regulation of microtubule assembly [48,49]. However, the cytosolic localization of Alsin, the absence of observed activity on Ran GTPase, and the non-conservation of the 25 residues of RCC1 protein responsible for Ran-binding make it rather unlikely for the Alsin RLD domain to exert an activity on Ran. Additionally, although a wide variety of proteins display RCC1 domains, only RCC1 itself has demonstrated Ran guanine nucleotide-exchange activity [41,50]. Even though this issue is still under debate, it seems that the Alsin RLD domain may drive subcellular localization and endosomal association through protein–protein interactions [42,51].

A short variant of Alsin that only contains part of the RLD domain exists (Figure 1); hypotheses were made about the role of this short variant in the variable severity of clinical phenotypes and the protective role of lower motor neurons (LMN) from degeneration, but this latter speculation was not confirmed since although a patient carried both Alsin variants, he did not show signs of LMN involvement [42,43,51]. Further studies demonstrated that the predicted protein derived from the short form of ALS2, together with the disease-associated mutant forms, are intrinsically unstable and may, therefore, undergo proteasome-mediated degradation in cultured human cells, including cells sampled from a patient with ALS2 mutations [42].

#### 3.1.2. DH and PH Domains

Another two structured regions, which are almost invariably present together in homologues proteins, are the Dbl homology (DH) domain and the pleckstrin-homology (PH) domain. In general, DH domains form an eleven-α-helix bundle of ~200 residues, while PH domains are usually ~100 residue-modular domains composed of two perpendicular anti-parallel β-sheets, followed by a C-terminal amphipathic helix (Figure 1) [52]. Notably, the identification of the PH domain may not be an easy task owing to the versatility in lengths and sequence for loops connecting the β-strands. It was demonstrated that PH domains can bind phosphatidylinositol within biological membranes and proteins, such as the β/γ subunits of heterotrimeric G proteins [53], which belong to a family of proteins involved in signal transduction and amplification and act as molecular switches through their interaction with G-protein-coupled receptors (GPCRs). Another example of a protein-embedding phosphatidylinositol that can be bound by PH domains is the protein kinase C [54] that regulates numerous cellular responses such as gene expression, protein secretion, cell proliferation, and inflammatory response. These interactions allow PH domains to participate in the recruitment of proteins to different membranes to address them to appropriate cellular compartments or enable them to interact with other components of the signal transduction pathways. The fact that PH domains invariably exists in the presence of DH domains makes it reasonable to think that a functional interdependence exists between these domains. Moreover, it was demonstrated that DH/PH domains accommodate within the DH/PH domain junction interface [55,56] and, notably, that Rho proteins are involved in the organization of the actin cytoskeleton, in neuronal morphogenesis, and in various signaling cascades [40]. Previous in vivo and in vitro studies suggested that Alsin DH/PH domains catalyze GEF on both Rab5 and Rac1 [49], but although it was demonstrated that DH/PH domains directly and specifically interact with Rac1 [57], further tests on COS-7 and Hela cells showed that Rac1 GEF activity in these cells is not relevant [57].

#### 3.1.3. MORN and VPS9 Domain

The VPS9 domain is common to numerous mammalian proteins and, more precisely, to GDP-GTP, which is involved in the activation of Rab5, a protein belonging to the Rab protein family of signal-transducing GTPases that cycle between active GTP-bound and inactive GDP-bound forms. This domain is composed of about 140 residues and catalyzes nucleotide exchange on the Rab5 GTPase, a regulator of endocytosis and endosome formation [58]. From a structural point of view, the VPS9 domain forms a layered fold of six α-helices (Figure 1). Notably, in the Rabex5 VPS9 domain, which shares homology with the Alsin VPS9 domain, the second and third helices form a helical hairpin that supports the four other helices [59]. Mutations of conserved residues (Asp 313, Pro 317, Tyr354, and Tyr 357) from the fourth and sixth helices hampered GEF activity on Rab5 and Rab21, suggesting that these two helices and the loop at the N-terminus of the fourth helix are the preferential interaction sites for Rab5 and Rab21 GTPases on the VPS9 domain [59].

The MORN motif is a tandem array of eight membrane occupation and recognition nexus motifs of 23 amino acids each [40] found in repeats in several proteins (Figure 1). Its function is unknown, but the repeats suggest involvement in Rab5-GEF activity through an association of Alsin protein with intracellular membranes [24]. Alsin MORN motifs and the VPS9 domain seem to work in synergy to drive endosome enlargement: in fact, a study on truncated variants showed that the lack of either the MORN motifs or the VPS9 domain abolished the GEF activity on Rab5 in vitro, whereas RLD-truncated variants promoted the formation of enlarged endosomes [41]. Moreover, the MORN/VPS9 domains are involved in Alsin self-interactions. Experimental evidence, using truncated variants in yeast cells, showed that the 1233-1351 and 1351-1454 regions are indispensable for Alsin homo-oligomerization. Additionally, it showed that although Rab5 binding to MORN/VPS9 is maintained if Alsin homo-association is hampered, the GEF activity on Rab5 is abolished [44].

The C-terminal half, including the MORN/VPS9 domains, is also involved in the endosomal localization of ALS2, while RLD in the long form of Alsin, after Rac1 binding, has an inhibitory role preventing ALS2 from being re-sequestrated by the self-interaction with RLD and MORN/VPS9 domains due to interaction with membrane ruffles [57]. It was found that the N-terminus with truncated ALS2, which therefore only contained the MORN/VPS9 region, was still capable of homo-dimerizing or homo-oligomerizing in mammalian cells, demonstrating that the C-terminal region is essential for Alsin homo-association [57].

### 3.2. Alsin Mutations Lead to Signalling Pathways Alteration

As clarified in the previous sections, ALS2 mutations reverberate in changes in the sequence of the Alsin protein. Such sequence modifications could impact the structural organization of the protein with consequential influence on intra-protein inter-domain interactions or protein–protein interactions of either Alsin with itself or Alsin with other protein partners. Changes in molecular interaction modes can result in changes in binding affinity or kinetics. Such changes can drastically alter protein cascades, strongly impacting cellular function. Although there are still not enough studies at the molecular level that can provide insight into the mechanisms of correlation between structure and function in Alsin, there are several experimental studies that have been able to assess Alsin’s misfunctions in the presence of specific molecular alterations, e.g., mutations. These studies are rationalized in the following sections.

#### 3.2.1. Alsin’s RLD Domain and Altered Trafficking of AMPA Receptors

One of the causes of the degeneration of motor neurons involved in IAHSP is glutamate-mediated excitotoxicity, a pathological process that leads to neuron damage and death upon over-activation of receptors; more precisely, the α-amino-3-hydroxy-5-methylisoxazole-4-propio-nate (AMPA) glutamate receptors. AMPA glutamate receptors are ligand-dependent ion channels, allowing ions such as N^+^, K^+^, Cl^-^, and Ca^2+^ to pass through the neuron membrane upon binding of a neurotransmitter [60]. AMPA glutamate receptors are homo- or hetero-tetramers consisting of a combination of GluR1, GluR2, GluR3, and GluR4 subunits. Notably, both GluR1 and GluR2 play an important role in synaptic plasticity. Moreover, neurons that are exempt from GluR2-containing receptors are more exposed to excitotoxicity. The GruR2 subunit is usually located on the synaptic surface owing to its interaction with the Glutamate receptor-interacting protein 1 (GRIP1), responsible for GluR2 transport to dendrites in both postsynaptic membrane and intracellular compartments, to form the AMPA-glutamate receptor [51,61,62]. GRIP1 interacts with Alsin’s N-terminal RLD domain both in vitro and in vivo [51,61]. This is true not only for the long form of Alsin but also for the short form, which contains a partial RLD domain and also interacts with GRIP1, although in a weaker fashion [61]. A decrease in GluR2 and altered distribution of GRIP1 is observed upon the loss of Alsin. [51,61,62]. The experimental data, therefore, suggest that Alsin might modulate the traffic of AMPA receptors thanks to its interaction with GRIP1, and also that it may play a neuroprotective role against excitotoxicity and, therefore, this pathway may be envisioned as a potential therapeutic target for the treatment of Alsin and motor neuron pathologies [40,51,61,62].

#### 3.2.2. Alsin Neuroprotective Role against SOD1 Mutations

It is firmly established that the mutations on chromosome 21q22.1 encoding for the Cu/Zn-superoxide dismutase (SOD1) can induce the degeneration of motor neurons, since it has been shown that mutated SOD1 acquires toxic properties [40,63,64,65]. SOD1 is a very abundant protein within the central nervous system; it constitutes 1% of all brain proteins [64,66]. Although the effect of SOD1 mutation is not yet fully understood, it is known that it leads to oxidative stress, mitochondrial dysfunction, excitotoxicity, protein aggregation, and inflammation. These alterations are not mutually exclusive and they may contribute to motor neuron degeneration [64]. Overexpression of Alsin may play a neuroprotective role in motor neurons by inhibiting the toxicity induced by SOD1 mutations through direct interaction between Alsin and mutated SOD1 [40]. Interestingly the DH/PH domain plays a dominant role in the above-mentioned interaction. It is worth noting that the above-highlighted Alsin–SOD1 interaction does not occur in the presence of wild-type SOD1 [40,64,65,67,68,69]. Moreover, ASL2 and SOD1 may share some functional characteristics. This is suggested by recent studies on genetically modified mice (ALS2-mutated mice and SOD1-mutated mice) showing altered toxicity due to inhibitors of mitochondrial complex 1 [63,67]. In a greater detail, transgenic mice endowed with Alsin-deprived neurons have been shown to be more sensitive to oxidative stress and to cell death induced by paraquat, an herbicide that is extremely toxic for cells. Taken together, the findings suggest that although the absence of Alsin may not be the sole factor inducing neurodegeneration in mice, it causes enhanced sensitivity to oxidative stress [51,68]. A clear neuroprotective role of Alsin against the degeneration of motor neurons induced by SOD1 mutations has not yet been demonstrated in vivo [63,68]. Therefore, a neuroprotective role has been established for Alsin in the presence of mutated SOD1 but not in the case of other neurotoxic insults [65,69]. Further evidence in favor of Alsin’s neuroprotective role is the observation that mutated ALS2 and SOD1 are co-localized, which was made by means of immunohistochemical analyses on transgenic rodents [65].

#### 3.2.3. Alsin DH/PH Stimulates Rac1-PAK Binding and Neuron Growth

Interestingly, Alsin colocalizes with Filamentous actin (F-actin) inside the ruffles of the growth cone membrane and on F-actin-coated cone vesicles [13,57,70,71]. A recent study highlighted how Alsin’s DH/PH domain is essential for the activation of signaling pathways related to (i) GTPases such as Rho, and Rac1, and (ii) for p21-activated kinase (PAK1) [70]. The above-mentioned signaling pathways are known to promote neuron growth and development. Recent studies suggested that Alsin overexpression does not affect the development of the axons and dendrites of neurons. Moreover, experiments also showed how the length of neurites in wild-type mice is significantly increased, while in mice endowed with DH domain-mutated Alsin, neurite growth is not altered [57,70].

#### 3.2.4. Alsin VPS9: Endosomal Trafficking and Rab5-Mediated Mechanisms

Endosomal trafficking is a recurrently affected pathway in several neurodegenerative diseases. In the following, we describe a number of sources of evidence that point towards a role of Alsin in the process involving Rab5 and suggest that the absence of Alsin or its mutations affect endosomal trafficking at different levels. Neurons use endocytosis to communicate with each other or with other tissues and to survive [62]. Several in vitro studies support the fact that Alsin dysfunction affects endosomal trafficking owing to the alteration of the VPS9 domain Rab5-mediated pathway [11,13,41,42,62,71,72]. Rab5 plays a fundamental role in the regulation of organelle binding, endosomal fusion, and motility in endocytosis. To prove this, experiments were performed where whole Alsin and truncated forms were found to be expressed [51]. It was observed that all forms were capable of triggering the release of GDP from Rab5, leading to endosomal budding [51]. Furthermore, overexpression of full-length Alsin results in a less efficient stimulation of endosomal fusion by Rab5 compared to RLD-domain-truncated Alsin, suggesting that the RLD domain can prevent Alsin from associating with early endosomes, thus acting as a negative regulator of Rab5-mediated endosomal fusion [13,41,51,62]. Interestingly, in ALS2 knockout homozygous (ALS2^(−;−)^) mouse neurons, an excessive amount of positive Rab5 vesicles and enlarged endosomes were observed, accompanied by a substantial decrease in endosomal motility [51,62]. Moreover, ALS2^(−;−)^ mice neurons displayed reduced cell bodies and axons compared to those of wild-type ones; similar effects were observed regarding cerebellar weight and total brain weight [71]. This evidence would support the idea that malfunctioning or absent Alsin may cause a blockage or slowing of the endosomal trafficking of neurotrophic receptors and lead to decreased neurotrophic support. Even the primary neurons’ function was shown to be altered due to Alsin dysfunction since ALS2^(−;−)^ mice suffer from disturbance of the endosomal transport of insulin-like growth factor (IGF1) and brain-derived neurotrophic factor receptors [41,51,62,72]. Moreover, Alsin is gathered inside the membranous compartments during micropinocytosis thanks to the activation of Rac1 [13,57,71], supporting the role of Alsin in mediating the fusion between early endosomes and macropinosomes. More specifically, the RLD domain drives the localization of Alsin and, therefore, it is essential for the recycling of Alsin in membranes [41,42,71]. Furthermore, missense mutants, despite displaying Rab5-GEF activity in vitro, are unable to localize Alsin correctly within early endosomes and/or macropinosomes even though the Rac1-induced micropinocytosis are active [57]. All the missense mutants passed from the cytosol to membrane ruffles, but they did not reach the endosomes or macropinosomes [13]. This observation suggests that the homo-tetrameric structure of Alsin may be required to activate this pathway; this means that a disordered Alsin structure leads to the loss of normal cellular functions [13]. Further support for the involvement of Alsin in endolysosomal trafficking is given by experimental evidence of Alsin colocalization with p62/LC3-positive autophagosomes [73]. 

In light of this, it has been suggested that Alsin may play a role in the maturation of autophagosomes into late endosomes and eventually participate in lysosome formation by accompanying Rab7 recruitment.

In conclusion, it is not yet possible to establish a comprehensive and consistent picture of the specific activity of Alsin in endosomal trafficking, but the data reported above unequivocally show that the absence of Alsin or certain mutations drive toward a plethora of effects in this process.

## 4. Population View on ALS2-Related Pathologies

As stated previously, given the low prevalence of these pathologies, studies in the literature usually account for a few cases, mostly related to one or two families. Therefore, this work tried to retrieve, collect, and summarize the information on IAHSP, JPLS, and JALS available in the literature to date. Without any presumption of providing a conclusive answer to the question that seeks to identify clear associations between the type of mutation on the ALS2 gene, and the effects on Alsin structure–function relationships, protein–protein interactions, cell malfunction, and the onset/development of MND disease, this review section points toward a rational collection of known ALS2 mutations, protein expression types, disease types, and symptoms that may aid future researchers in retrieving relevant literature and data for specific purposes. The collected data are comprehensively reported in the relevant tables in the (Appendix A).

Concerning IAHSP, Appendix A report interlinked information concerning patients in terms of mutation type, pathology onset, and symptoms taken from literature studies detailed in the following. Helal et al. reported three Iranian-originated families with a total of 11 children affected by IAHSP born to healthy consanguineous parents. It was noted that within the same family with the same genotype (c.1640 + 1G > A), there is a difference in the expressed phenotype. This result was taken as a starting point to analyze the influence of environmental factors or epigenetic factors on the variability of symptoms [4]. Daud et al., focused on two other families for which they reported specific mutations of ALS2: the first mutation is the nonsense c.2998delA (p.Ile1000*), responsible for the interruption of ALS2 transcription after ~1000 residues, and the second one is the c.194T > C (p.Phe65Ser) missense mutation. Unlike the previous study described above, a great phenotypic homogeneity was found [1], which highlighted the fact that further research is still needed to understand the factors within the genome that can lead to a different phenotypic manifestation. Eymard-Pierre et al. and Sprude et al. reported the same mutation, c.470G > A (p.Cys157Tyr), of ALS2 in two unrelated Turkish families [74,75]. The first study focused on two sisters, daughters of consanguineous parents who started walking with support at age 3 and 7 and lost ambulatory skills at age 12 and 10 (family 8; Appendix A) [74]. On the other hand, the second study referred to a child who started walking at 14 months without any support and could still walk at 11 years old (family 25; Appendix A) [75]. These two studies further highlight the phenotypic difference that exists despite the genotypic homogeneity, showing how, in the second case, the mutation had a less aggressive course.

Concerning JPLS, Appendix A report interlinked information concerning patients in terms of mutation type, pathology onset, and symptoms taken from literature studies detailed in the following. Previous investigations highlighted the importance of the C-terminal domain, which, in Alsin, is endowed with a GEF activity and which lack can cause JPLS or IAHSP. These results showed how important this domain is and how the loss of that functionality can cause these diseases. Moreover, researchers noted that the N-terminal domain performs a structural function, and its lack causes a loss of stability of the protein as reported in another study [2,49,74,76,77]. This agrees with what other researchers reported in another study; namely, they noted that most pathogenic ALS2 mutations cause the production of a truncated Alsin protein lacking the C-terminal, which supports the hypothesis that the lack of the GEF activity could be one of the primary causes of diseases related to the ALS2 gene, although affected oligomerization cannot be excluded [17,24,44]. Mintchev and collaborators showed that mutations within the introns of the ALS2 gene can also cause incorrect transcriptions of the Alsin protein. In the reported case, the mutation c.2980-2A > G located in the splice acceptor site of intron 17 caused at least the absence of exon 18 in the mRNA sequence and it is possible that it also caused a premature stop in exon 19, probably giving rise to a truncated protein without the C-terminal region [17]. Moreover, an interesting case study emerges from two unrelated patients, showing the same mutation, namely c.4573dupG and the corresponding protein with frameshift mutation p.Val1525GlyfsTer17, [75,78]. Interestingly, these two patients were diagnosed with two different pathologies, IAHSP and JALS. The two children, in fact, despite having the same mutation, showed different phenotypes. A case reported by Sprute et al. depicts an IAHSP patient, with the age of onset at 16 months, and a subject who was never able to walk [75]. In the other study reported by Sheerin and colleagues, symptoms were described as a form of JALS. Indeed, the onset of symptoms was at 2–3 years and the patient lost the ability to walk at 8 years [78]. Furthermore, both showed dysarthria and dysphagia, but with different severities [75,78].

Concerning JALS, Appendix A report interlinked information concerning patients in terms of mutation type, pathology onset, and symptoms taken from literature studies detailed in the following. In all the reported cases, spasticity of the limbs, but not bulbar involvement, was noted as the initial hallmark, in contrast to what is more often observed. Another interesting aspect is JALS related to mutations in the FUS gene, which cause disease onset at a later age, on average 18 years, compared to those due to mutations in ALS2 genes, which occur at 6 years on average. The situation is different for the lifespan factor, which in the case of FUS mutation is 12 months, whereas, for mutations of the other involved genes, it can reach 150 months or more [28,79,80,81,82].

Starting with the data obtained from patients and reported in Appendix A, it is also possible to assess the frequency of mutation types of ALS2 related to pathological conditions and their distribution over Alsin’s structured domains. Table 2 highlights the distribution of ALS2 frameshift, missense, or nonsense mutations in the three considered MNDs. Interestingly, based on the collected data, IAHSP can be correlated to almost 30 different mutations (Frameshift, Missense, and Nonsense), in contrast to the other two diseases, which, based on known cases, are correlated to a smaller number of mutations. It is also interesting to note that most of the mutations are in the RLD domain (Figure 2 and Table 3), which has already been highlighted by experimental studies as a crucial domain for Alsin’s functional development [13]. Nevertheless, the fact that the same pathology can be triggered by mutations on different domains of the protein might also suggest that interdomain interactions and self-assembly mechanisms are important for the performance of the protein’s functions.

Nonetheless, with the current available knowledge on MNDs’ patient features, it is not possible to draw any definitive conclusions or a direct correlation between a specific mutation and the rate of disease progression or the severity of symptoms of any of the three diseases of interest discussed in this review. The need for further comparative studies and, most importantly, additional data remains open.

## 5. Conclusions and Future Perspectives

We reviewed recent literature concerning ALS2-related neurodegenerative diseases, i.e., IAHSP, JPLS, and JALS, by employing a comparative multilevel approach. The macroscopic-level investigation helped to classify their clinical features such as age at onset, symptoms, and hallmarks. At the microscopic scale, a comprehensive review, concerning Alsin’s (i) structural biology, (ii) related protein networks, and (iii) alteration-driven aberrant behavior, was carried out. Lastly, a population level view was adopted in an attempt to organize the sparse literature on reported clinical cases with attention to age, geographic distribution, proteins, gene mutations, and other relevant variables. Our approach stresses the complexity of Alsin-related neurodegenerative pathologies, in which a mutation on the same protein may result in a wide range of clinical symptoms, or the same genetic insult may lead to distinct diseases [75,78]. Further studies should focus on Alsin’s structure–function relationships, which may help both in terms of understanding the molecular defects underlying these pathologies and in the discovery of possible therapeutic approaches. In this respect, a structural and biochemical approach should also address the identification and characterization of partners and interactors. Experimental methodologies including X-ray crystallography, nuclear magnetic resonance, and even cryo-electron microscopy, supported by in silico techniques such as homology modelling, molecular docking, and molecular dynamics, will shed light on the protein’s conformational dynamics and interactions in the oligomerization process. So far, an homology model for the RLD was developed [46], together with a very recent homology model and molecular dynamics investigation of Alsin’s DH/PH domain [83]. Instead, developing molecular models of all of Alsin’s atoms is a milestone in the comprehension of ALS2-related pathologies. The systematic identification of sub-cellular localization in normal and pathological conditions is also a fundamental requisite to clarify the role of Alsin in neuronal health and disease. Moreover, another crucial point regarding rare Alsin-related diseases pertains to the availability of information, which, at present, is limited in terms of the amount, availability, heterogeneity, dispersity, and fragmentation of patient data. More efforts should be oriented toward providing free tools for data sharing at all levels, from more basic science to more applied clinical levels.

## Figures and Tables

**Figure 1 biology-11-00077-f001:**
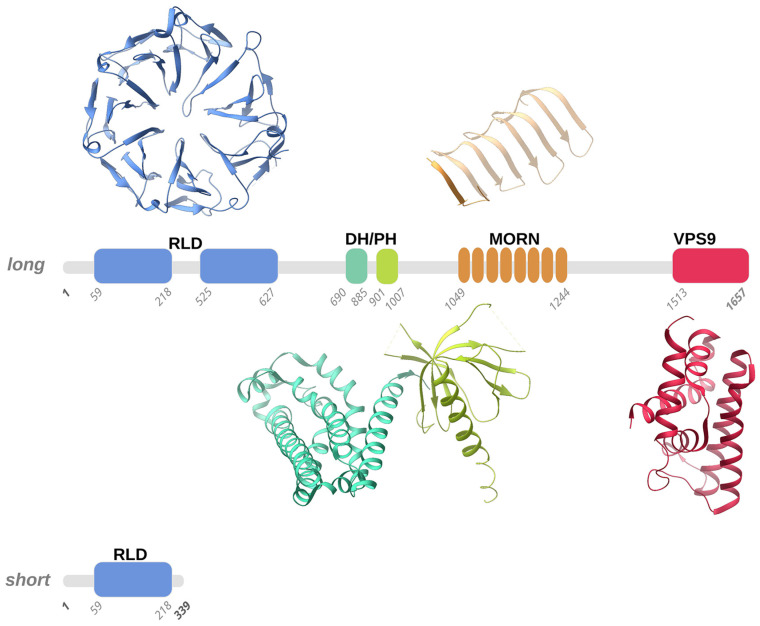
Schematic representation of the long and short forms of human Alsin. According to homology modelling, four structured domains, numbered according to the Uniprot Q96Q42 entry, were predicted in human Alsin, namely the RLD (RCC1-like domain), DH-PH (Dbl homology-pleckstrin homology), MORN (membrane occupation and recognition nexus), and VPS9 (vacuolar protein-sorting 9) domains. We used representative structures of these domains (PDB entries 1A12, 2Z0Q, 6T4D, and 2OT3) to illustrate the 3D folds that the RLD, DH-PH, MORN, and VPS9 domains predicted in human Alsin, respectively, may adopt. Structured domains are shown as ribbons with the same color code as the one utilized for the sequence schematic representation.

**Figure 2 biology-11-00077-f002:**
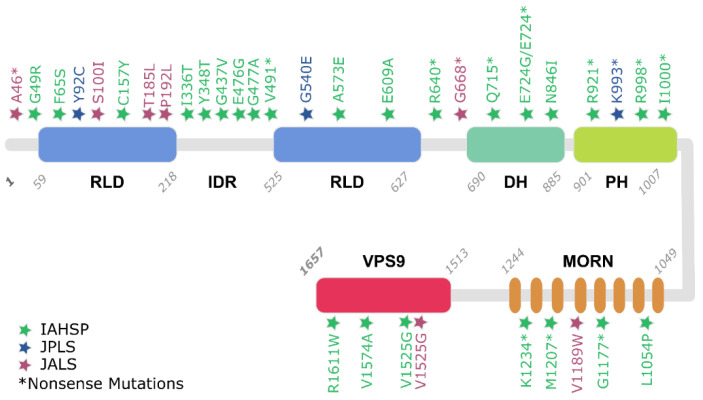
Reported Alsin mutations, inducing IAHSP, JPLS, and JALS. Mutation positions are indicated with colored stars and type of mutation is reported following a commonly used nomenclature (e.g., G49R).

**Table 1 biology-11-00077-t001:** Comparison of the main clinical features characterizing ALS2 gene mutation-related diseases, i.e., IAHSP, JPLS, and JALS.

	IAHSP	JPLS	JALS
Inheritance	Autosomal recessive	Autosomal recessive	Autosomal recessive
Age of onset	1–3 years old	1–3 years old	4–8 years old
Life expectancy	Adulthood	Adulthood	7 months to 17 years old
Genetic causes	ALS2 mutation	ALS2 mutation	ALS2 mutation (Other forms are caused by SETX, FUS, UBQLN2, SPG11, SIGMAR1)
Neuron alterations	Degeneration of both upper and lower motor neurons	Progressive degeneration, upper motor neurons	Degeneration of both upper and lower motor neurons
Symptoms	Lower limb weakness and spasticity progressing towards quadriplegia, wheelchair dependence by the age of 10, followed by tetraparesis, feeding dependence on gastrostomy	Lower limb weakness and spasticity, wheelchair dependence by adolescence, motor speech impairment, saccadic eye movements	Lower limb weakness and spasticity, face muscle spasticity, bladder dysfunction, dysarthria, sensory disturbances, and sometimes mental retardation and sclerosis

**Table 2 biology-11-00077-t002:** Frequency of mutation types of ALS2 related to pathological conditions.

Disease	Frameshift	Missense	Nonsense
IAHSP	12.	6	8
JPLS	2	1	0
JALS	4	2	1
%TOT	50%	25%	25%

IAHSP, Infantile-onset Ascending Hereditary Spastic Paralysis; JPLS, Juvenile Primary Lateral Sclerosis; JALS, Juvenile Amyotrophic Lateral Sclerosis; %TOT, percentage over the total of cases.

**Table 3 biology-11-00077-t003:** Frequency of mutations divided into different predicted domains.

Disease	RLD	DH/PH	MORN	VPS9
IAHSP	12	7	4	3
JPLS	2	1	0	0
JALS	4	1	1	1
%TOT	50%	25%	14%	11%

IAHSP, Infantile-onset Ascending Hereditary Spastic Paralysis; JPLS, Juvenile Primary Lateral Sclerosis; JALS, Juvenile Amyotrophic Lateral Sclerosis; %TOT, percentage over the total of cases.

## Data Availability

Not applicable.

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
