# Peer review of "ALS2-Related Motor Neuron Diseases: From Symptoms to Molecules"

_biology, 2022, doi:10.3390/biology11010077_

Round 1

Reviewer 1 Report

The review is logically set out and is worth reading. This article describes ALS2 related motor neuron diseases from several perspectives including clinical characteristics, especially the characteristics of the affected population, the structure and function of ALSin protein, and the possible pathogenesis at the molecular level.

Questions about Part2

  1. The age of onset of JPLS was described in Part 2.2 as “before the age of 25”, but in Table 1, the age of onset is “1-3 years old”.
  2. The life expectancy of JALS was described in Part 2.3 as “Death usually occurs upon respiratory failure between 7 and 84 months from the onset”, which does not correspond to the age of onset and life expectancy in Table 1.
  3. “Finally, JALS appears to be the most severe phenotype from studies, while JPLS the mildest. Severity is related to the lifespan: individuals with IAHSP and JPLS may expect to have an over 40-year life expectancy, while those with JALS have a shorter lifespan often owing to respiratory failure” This conclusion is not rigorous. According to the content of this paper, the clinical manifestations of IAHSP seem more serious than that of JPLS.

Suggestions

  1. in Part 4,population view on ALS2 related pathologies, it will be better if the author can describe whether mutations in different domains of this gene will affect the severity of the disease.
  2. In Fig.2, suggest the author the site number to amino acid mutational style. For example, 65 changes to F65S.

Author Response

Dear Reviewer, please find all the answers in the attached file

Reviewer 2 Report

Overall, this review article is a good overview of ALS2 pathobiology but it is rudimentary.  There are no significant insights into how mutations in alsin can lead to three distinct motor neuron disease phenotypes. It is unclear if this review comprehensive covers the literature.

The authors need to provide a more comprehensive table listing all of the currently known alsin mutations linked to IAHSP, JPLS and JALS. While Figure 2 does provide some of this information, there needs to be more specific information such as the type of mutation and changes in the amino acid sequence.

This reviewer agrees with the opinion of the authors that more structural and biochemical characterization of alsin mutations is needed. The authors should explore this need in greater detail.

From a stylistics perspective, the text of the manuscript needs to be more cohesive. At present, it reads as a list of sentences without any transitions. Also, some of the wording in the introduction and conclusion is superfluous.

Author Response

(The authors gave the same response as above.)

Reviewer 3 Report

In their review titled " ALS2 related motor neuron diseases: from symptoms to molecules ", the authors provide a clear, and concise overview of the importance of ALSIN protein in mechanistic and disease perspective. The authors described well the clinical and molecular differences and similarities between Infantile-onset ascending hereditary spastic paralysis (IAHSP), Juvenile Primary Lateral Sclerosis (JPLS) and Juvenile Amyotrophic Lateral Sclerosis (JALS) delivered a generally even critical analysis of the literature. 

Author Response

(The authors gave the same response as above.)

Reviewer 4 Report

This manuscript reviews the knowledge about the impact of mutations in the ALS2 gene in the pathogenesis of three motor neuron diseases (Infantile-onset Ascending Hereditary Spastic Paralysis, Juvenile Primary Lateral Sclerosis and Juvenile Amyotrophic Lateral Sclerosis).

The paper highlights well the overalps and divergences among the three motor neuron diseases by correlating clinicals signs to molecular alterations at protein level.

The review is well structured, and opinions are properly argumented.
The figures, tables and Supporting Information (reporting known mutations) are clear, and effective in illustrating the main concepts.

The only suggestion is to add a comment about co-localization of ALS2 with LC3 and p62, further supporting ALS2 implication in endolysosomal trafficking through the fusion between endosomes and autophagosomes (https://doi.org/10.1371/journal.pone.0009805).

Author Response

(The authors gave the same response as above.)
